# Factors associated with non-adherence to Antiretroviral Therapy (ART) in adult individuals living with Human Immunodeficiency Virus (HIV) in Mogadishu, Somalia: A hospital-based cross-sectional study

Abdirahman Khalif Mohamud[1]*, Bashiru Garba[1,2], Najib Isse Dirie[3], Pamornsri Inchon[4], Maryan Abdullahi Sh Nur[5], Jamal Hassan Mohamoud[6], Mohamed Mustaf Ahmed[1]

1 Faculty of Medicine, and Health Science, SIMAD University, Mogadishu, Somalia, 2 Faculty of Veterinary Medicine, Usmanu Danfodiyo University Sokoto, Sokoto, Nigeria, 3 Department of Urology, Dr. Sumait Hospital, Faculty of Medicine, and Health Sciences, SIMAD University, Mogadishu, Somalia, 4 Department of Public Health, School of Health Science, Mae Fah Luang University, Chiang Rai, Thailand, 5 Department of Obstetrics and Gynecology, Dr Sumait Hospital, Faculty of Medicine, and Health Sciences, SIMAD University, Mogadishu, Somalia, 6 Department of Public Health, Faculty of Medicine, and Health Sciences, SIMAD University, Mogadishu, Somalia

* Aabihaaji@gmail.com

## Abstract

### Background

Antiretroviral Therapy is an effective method against HIV, reducing mortality and opportunistic infections. In Somalia, high rates of these infections and low adherence to therapy have been reported. However, factors contributing to non-adherence remain unidentified. Therefore, The study aims to assess the magnitude of antiretroviral therapy non-adherence and identify associated factors for individuals living with HIV in Mogadishu, Somalia.

### Methodology

A hospital-based cross-sectional study was conducted using a questionnaire containing socio-demographic, behavioral, familial, psychosocial, clinical, and laboratory characteristics. Antiretroviral Therapy adherence was assessed by quantifying the number of prescribed pills taken in the past month. Those who adhered to ≥95% of the prescribed Antiretroviral Therapy drugs were considered adherent to Antiretroviral Therapy. Logistic regression was used to identify factors associated with Antiretroviral Therapy non-adherence at significance levels of p < 0.05.

### Results

Out of 453 participants, the magnitude of non-adherence to ART among people living with HIV in Mogadishu, Somalia, was 138 (30.5%) with its 95% CI: 26.3–34.9. The

**Data availability statement:** All relevant data are within the manuscript and its Supporting Information files.

**Funding:** This study received funding from SIMAD University through the Center for Research and Development under grant number 2023/SU-CRD/FMHS/P009.

**Competing interests:** All authors declare that they do not have any competing interests.

**Abbreviation:** ART; antiretroviral therapy, HIV; human immunodeficiency virus, OIs; opportunistic infections, HAART; high active antiretroviral therapy.

predominant reasons for missing ART were forgetting (7.3%), transportation issues (5.5%), side effects and being busy with other commitments (4.4%), stigma (3.8%), and ART facilities being distant from home (2.6%). In multivariable logistic regression, living alone (AOR=1.81, 95%CI:1.182–2.782), perceived stigma (AOR=3.19, 95%CI:1.784–5.720), smoking cigarettes (AOR=2.96, 95%CI:1.759–5.003), and living with a co-morbid chronic disease (AOR=1.80, 95%CI:1.155–2.813) were factors associated with ART non-adherence among people living with HIV in Mogadishu, Somalia.

## Conclusion

Individuals living with HIV in Mogadishu, Somalia suffer sub-optimal adherence to ART. Priority should be given to strategically addressing the needs of those living alone and individuals with co-morbid chronic conditions, eliminating HIV stigma, and discouraging substance abuse.

## Background

Human Immunodeficiency Virus 1 and 2 (HIV-1 and HIV-2) contribute to a worldwide public health challenge. Sub-Saharan Africa bears the highest burden, with a population exceeding 25.7 million individuals living with HIV [1]. Antiretroviral Therapy (ART) represents the sole efficacious approach in the battle against HIV due to its crucial role in maintaining viral suppression, restoring immune function, and reducing mortality and the incidence of opportunistic infections(OIs). In addition, it enhances the overall quality of life and life expectancy among people living with HIV [2,3]. Achieving a minimum of 95% adherence to Antiretroviral Therapy (ART) is considered a hallmark of successful HIV treatment, capable of leading to effective virus suppression [2,3]. Effective HIV treatment can be challenging without the client's perfect ART adherence [4].

Sub-Saharan African nations continue to face a heightened susceptibility to suboptimal adherence to ART, primarily attributed to several factors, including the region's status as the continent most severely affected by HIV, its inadequate public health infrastructure, and the healthcare shortage of professionals [5,6]. An average of 72.9% of ART adherence is reported in sub-Saharan Africa [5]. An African meta-analysis reported a wide range of ART adherence rates, from 32.9% to as high as 94%, which is under the acceptable standard [6]. Reported factors associated with ART non-adherence include non-disclosure of HIV status, marital status, advanced HIV stage, lack of social support, low socioeconomic status, stigma and discrimination, distance to ART facilities, limited treatment accessibility, technological inadequacies, ART side effects, unemployment, depression, mental health issues, and substance abuse [7–12]. Inadequate ART adherence has far-reaching consequences, including developing drug resistance, elevated mortality rates, suboptimal treatment outcomes, and a heightened prevalence of opportunistic infections [13].

Healthcare coverage for people living with HIV in Somalia is inadequate, with limited ART facilities available [14]. They face widespread stigma, leading to self-isolation and limited life opportunities. The HIV response in Somalia is jeopardized by discrimination with inadequate social support [14,15]. In addition, clinics and previous Somali studies reported a high incidence of opportunistic infections due to inadequate adherence to ART [16]. A high prevalence of depression linked to insufficient ART adherence has been reported among individuals with HIV in Somalia [17]. However, no previous Somali studies have identified the factors contributing to ART non-adherence. Meanwhile, Somali healthcare providers and policymakers have

directed their efforts toward establishing an effective mechanism to improve ART adherence among this vulnerable population. These efforts become worthless without accessing determinants of ART non-adherence, as this aspect has yet to be previously reported in Somalia. Up-to-date information in this regard is vital for enhancing HIV treatments, interventions, and strategies at the national and international levels. Hence, this study aims to assess the magnitude of ART non-adherence and identify its associated factors among individuals living with Human Immunodeficiency Virus (HIV) in Mogadishu, Somalia.

## Method

### Study design

A hospital-based cross-sectional study was implemented from May to August 2023 to assess the magnitude of non-adherence to Anti-retroviral Therapy (ART) and identify factors associated with among people living with HIV in Mogadishu, Somalia.

### Study area

The study was conducted at Banadir Hospital in Mogadishu, Somalia. Mogadishu, the capital of Somalia, is the largest and most densely populated city in the country. The hospital serves as a prominent teaching and referral institution, standing as the largest healthcare facility in the city and being one of the two publicly operated hospitals under the administration of the Somali Government. Furthermore, the study hospital hosts the largest Antiretroviral Therapy (ART) facility in southern and central Somalia. Over 70% of individuals living with HIV in the country seek care at this facility and have a recorded file [16–18].

### Study population and eligibility criteria

The study population included individuals diagnosed with HIV infection who were undergoing anti-retroviral therapy (ART) for at least the last six months, had medical records at the ART unit of the study hospital, and were physically present during the data collection period. Individuals who were not mentally capable of participation, unwilling to participate, unable to communicate verbally, had hearing impairments, had baseline incomplete medical record files (CD4 (cell/mm$^3$), HB (g/dl), HIV Viral load (copies/mL), history of co-morbid disease, and history of opportunistic infections), or declined to provide consent were excluded from this study.

### Sample size and sampling techniques

A standardized formula for cross-sectional studies was used to determine the required sample size [19]. The formula used was n = $Z^2_{\alpha/2}$ P (1 - P)/d², where $Z^2_{\alpha/2}$ equated to 1.96, P represents the prevalence proportion, which is 37.78%, as obtained from a similar study conducted in Cameroon [20], and d being the desired precision level of 0.05. Accordingly, our study required a total sample size of 398 respondents, including 10% non-response. However, this study recruited a total of 453 participants. A systematic sampling technique was used to select participants.

### Research instrument

A well-structured, reliable, validated questionnaire was developed from an extensive literature review and discussed by three field experts [2–5,7,9–11,15,21]. The tool was initially developed in English, and then a language expert did forward-backwards translations to ensure consistency. Data was collected using the Somali version of the tool.

Content validity was assessed using the Item Objective Congruence (IOC) method [22] by three external experts (specialists in Infectious Diseases, senior ART Medical doctors, and Clinical Epidemiologist). Consequently, a 30-response pilot study with similar characteristics was conducted to enhance the questionnaire reliability and respondent understanding, and a satisfactory 0.81 level of Cronbach's Coefficient alpha values was achieved.

The outcome variable was Anti-retroviral therapy adherence, assessed by quantifying the number of pills remembered to have been taken in the last month from the prescribed medication. Patients who adhered to ≥95% of the prescribed Anti-retroviral medication were categorized as adherent to Anti-retroviral therapy, while those with adherence rates below 95% were classified as non-adherent [16,17,21,23,24].

The questionnaire contained: **i)** Socio-demographic characteristics, including respondents' age, gender, marital status, place of residence, educational level, occupation, and income. **ii)** Behavioral, familial, and psychosocial characteristics, including cohabitation status, perceived stigma, level of social support, disclosure of HIV status, proximity to the ART facility, mode of transportation to the ART facility, use of ART reminders, presence of supportive family members, cigarette smoking, engagement in multiple sexual partnerships, intravenous drug use, and khat consumption. **iii)** Clinical and laboratory characteristics, including level of CD4, baseline viral load, Hemoglobin (HB) levels, HIV stage, duration of living with HIV and receiving ART, regimen line, presence of ART side effects, history of missed ART appointments over the past four months, daily ART dosing, comorbid chronic diseases, and a history of opportunistic infections.

A 3-item social support scale was used, which was categorized as "poor support" for scores ranging from 3 to 8, "moderate support" for scores between 9 and 11, and "strong support" for scores from 12 to 14 [17,25]. Individuals diagnosed with diabetes, hypertension, and other chronic conditions are considered to have coexisting chronic diseases, whereas those without any history of such conditions are classified as having non-coexisting chronic illnesses. Eleven items on the HIV Stigma scale were utilized to assess perceived stigma related to HIV, and stigma variables were computed and categorized as either stigmatized or non-stigmatized, with the cutoff point set at the mean of the stigma variables [17,26,27].

## Data collection procedure

Three licensed ART medical doctors underwent a five-day training program to improve their familiarity with the questionnaire content. Subsequently, they verified the eligibility of potential study participants and provided information to eligible participants regarding the study objective. Upon obtaining consent, participants either signed a consent form or were fingerprinted (illiterate participants) to signify their participation agreement. Data collectors conducted individual face-to-face interviews and reviewed the case record files of each consenting participant following the research instrument in a confidential, private room that lasted approximately 20 minutes per participant.

## Data analysis procedure

Data was cleaned, coded, and entered on the spreadsheet imported into the SPSS version 20 (SPSS, Chicago, IL License) for analysis. Descriptive statistics were used for all categorical characteristics by presenting frequency with cross-tabulation, percentage, and p-value from Chi-square at <0.05 significant level. The continuous characteristics (Age, monthly income, and distance to the ART facility) were presented as mean with standard deviation (SD). The magnitude of ART adherence and non-adherence was presented frequency with a percentage and 95%CI. Univariable and Multivariable models of logistic regression analyses were

conducted to identify factors associated with ART non-adherence. In the univariable model, variables demonstrating a p-value less than 0.05 were chosen as candidates for inclusion in the multivariable model. The Hosmer-Lemeshow goodness of fit test was used to assess the goodness of fit of the final model [28]. Variables with p-values less than 0.05 were considered statistically significant.

Variance Inflation Factor (VIF) analysis was used to assess the potential existence of multicollinearity within the multiple regression analysis. A VIF value exceeding 10 is indicative presence of multicollinearity. However, the VIF values in this study ranged between 1.04 and 1.35 for all variables indicating the absence of multicollinearity among the variables in this study.

### Ethical approval and consideration of the study

This study followed the rules of the World Medical Association Declaration of Helsinki. Ethical approval was obtained from the Ethical Committee on Human Research at SIMAD University in Mogadishu, Somalia.

As protocol IRB number (2023/SU-IRB/FMHS/P0010) outlines, the Ethical Committee on Human Research at SIMAD University reviewed and approved the study protocol. Informed consent was obtained from all eligible participants, who were provided with a detailed explanation of the study's objectives and invited to participate voluntarily. The study objectives were informed to illiterate participants through their legally authorized representatives without coercion or undue influence. All study participants expressed their consent in writing or through fingerprinting (illiterate). Furthermore, participants were informed that they could join or discontinue the interview at any point. Confidentiality was kept, questionnaires were anonymous, and all data were presented anonymously.

## Results

### The magnitude of non-adherence to Antiretroviral Therapy (ART) in PLHIV

Out of 453 participants, the magnitude of non-adherence to Antiretroviral Therapy (ART) among people living with HIV in Mogadishu, Somalia, was 138(30.5%) with its 95%CI=26.3–34.9 (Table 1). The most common reasons for missing ART were simply forgetfulness (7.3%), due to transportation (5.5%), due to side effects and being busy with other things (4.4%), Stigma (3.8%), and ART facility away from home (2.6%) (Fig 1).

### Socio-demographic characteristics

Among the total of 453 individuals that participated in the study, 59.8% were male with a mean age of 31.68 (SD±8.145), with more than half (58.8%) being single, and 68.7% living in urban areas. The result 42.6% were employed with a mean income of 437.81 USD (SD±334.93) per month (Table 2).

Table 1. *Magnitude of non-adherence to Antiretroviral Therapy (ART) in PLHIV.*

| Characteristics | n (%) | 95%CI |
|---|---|---|
| **The magnitude of Non-Adherence to Antiretroviral Therapy (ART) Among HIV People** | | |
| Non-Adherence | 138 (30.5%) | 26.3-34.9 |
| Adherence | 315 (69.5%) | 65.1-73.7 |

## Reason missing ART

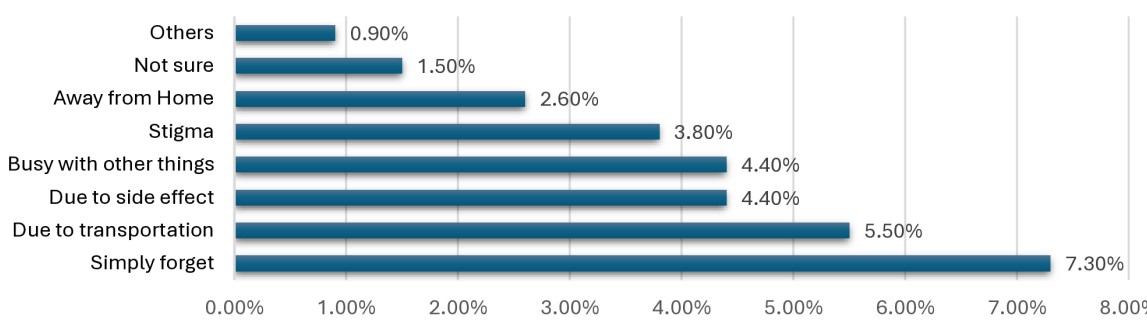

**Fig 1. Reasons for missing Antiretroviral Therapy (ART) among HIV people in Somalia (2023).**

**Table 2. Socio-demographic characteristics between adherence and non-adherence.**

| Characteristics | ART adherence | | Total n (%) | p-value |
|---|---|---|---|---|
| | Adherence n (%) | Non-adherence n (%) | | |
| **Age** | | | | 0.960 |
| 18-28 | 111(68.9%) | 50(31.1%) | 161 (35.5%) | |
| 29-38 | 146 (69.5%) | 64(30.5%) | 210 (46.4%) | |
| 38-60 | 58(70.7%) | 24(29.3%) | 82 (18.1%) | |
| Mean=31.68(SD±8.145) | | | | |
| **Sex** | | | | 0.177 |
| Male | 195(72%) | 76 (28.0%) | 271(59.8%) | |
| Female | 120 (65.9%) | 62 (34.1%) | 182(40.2%) | |
| **Marital status** | | | | 0.107 |
| Single | 171 (65.3%) | 91 (34.7%) | 262(58.8%) | |
| Married | 102 (76.2%) | 31 (23.3%) | 133(29.4%) | |
| Window | 30 (75.0%) | 10 (25.0%) | 40(8.8%) | |
| Divorced | 12 (66.7%) | 6 (33.3%) | 18(4.0%) | |
| **Resident** | | | | 0.782 |
| Urban | 215(69.1%) | 96 (30.9%) | 311(68.7%) | |
| Rural | 100 (70.4%) | 42 (29.6%) | 142 (31.3%) | |
| **Education** | | | | 0.549 |
| Illiterate | 93(67.4%) | 45(32.6%) | 138 (30.5%) | |
| Read or write | 84 (75.0%) | 28 (25.0%) | 112 (24.7%) | |
| Primary level | 93(67.9%) | 44 (32.1%) | 137 (30.2%) | |
| University level | 45(68.2%) | 21(31.8%) | 66 (14.6%) | |
| **Occupation** | | | | 0.243 |
| Employee | 142(73.6%) | 51(26.4%) | 193(42.6%) | |
| Businessperson | 79(68.1%) | 37(31.9%) | 116(25.6%) | |
| Jobless | 94(65.3%) | 50(34.7%) | 144(31.8%) | |
| **Monthly income (USD $)** | | | | 0.553 |
| 0–300 | 158 (69.6%) | 69 (30.4%) | 227 (50.1%) | |
| 301–600 | 81 (66.4%) | 41 (33.6%) | 122 (26.9%) | |
| > 601 | 76 (73.1%) | 28 (26.9%) | 104 (23.0%) | |
| Mean = 437.81USD (SD±334.93) | | | | |

### Behavioral, familial, and psychosocial characteristics of adherence and non-adherence

Most of the study participants 282(62.3%) cohabitated with people, 76.2% perceived stigma, 54.5% used ART remaining mechanism, 55.2% had a supportive family member, 18.3% smoked cigarettes, 23.2% chewed khat, and 19.2% were IV drug abusers (Table 3).

### Clinical and laboratory characteristics between adherence and non-adherence

Most of the study participants 352 (77.7%) had an HIV Viral load ≤1000 copies/mL, 19.2% experienced ART side effects for the past 4 months, 79.0% used ART ones daily, 58.3% had a co-morbid chronic disease, 24.7% had a history of opportunistic infections (Table 4).

### Factors associated with non-adherence to Anti-retroviral Therapy (ART) people live with HIV

In the Univariable logistic regression model, twelve (12) variables were significantly associated with non-adherence to ART at a significance level of 0.05. These variables were cohabitation status, perceived stigma, use of ART reminders, presence of supportive family members, cigarette smoking, intravenous drug use, khat consumption, HIV viral load, ART side effects, daily ART dosing, co-morbid chronic diseases, and history of opportunistic infections. These variables were selected as candidates for inclusion in the multivariable logistic regression model, and variables with p-values less than 0.05 were considered statistically significant and are presented in Table 5.

The odds of not adhering to Anti-retroviral therapy (ART) were 1.81 times higher (95%CI=1.182–2.782) for individuals living alone compared to those living with others. Those experiencing perceived stigma were 3.19 times more likely (95%CI=1.784–5.720) to exhibit non-adherence to ART than those not facing any stigma. Individuals who smoke cigarettes had a 2.96-fold increased likelihood (95%CI=1.759–5.003) of non-adherence to ART compared to non-smokers. Similarly, those with a co-morbid chronic disease were 1.80 times more likely (95%CI=1.155–2.813) to be non-adherent to Anti-retroviral therapy (ART) in contrast to those without such conditions (Table 5).

## Discussion

Anti-retroviral Therapy (ART) is the only practical approach for combatting HIV due to its role in sustaining viral suppression, restoring immune functionality, and diminishing mortality rates and opportunistic infections. Furthermore, it substantially enhances the overall quality of life for individuals living with HIV [2,3]. In this investigation, it was found that residents of Mogadishu, Somalia, who are living with Human Immunodeficiency Virus (HIV) suffer a burden of non-adherence to Anti-retroviral Therapy (ART), particularly those who have experienced stigma, cohabit alone, smoke cigarettes, and live with co-morbid chronic ailments. The predominant reasons for non-adherence to the prescribed ART doses include forgetfulness, transportation-related difficulties, adverse side effects linked to ART usage, preoccupation with other competing obligations, the pervasive stigma, and the inconvenience of accessing ART facilities that are distant from their residence.

The magnitude of non-adherence to ART observed in this study is remarkably higher when contrasted with prior research conducted in various African regions and a global context, such as Oromia, Ethiopia [3], Uganda [29], western Ethiopia [21], Togo, Ethiopia [30], Laos [31], Nairobi, Kenya [32], Southwest, Southeast, Northwest, and Eastern Ethiopia [33–36]. These

**Table 3. Behavioral, familial, and psychosocial characteristics of adherence and non-adherence.**

| Characteristics | ART adherence | | Total n (%) | p-value |
|---|---|---|---|---|
| | Adherence n (%) | Non-adherence n (%) | | |
| **Cohabitation status** | | | | 0.002* |
| Alone | 104 (60.8%) | 67(39.2%) | 171(37.7%) | |
| With people | 211(74.8%) | 71(25.2%) | 282(62.3%) | |
| **HIV Status Discloser** | | | | 0.058 |
| Yes | 101(63.9%) | 57(36.1%) | 158(34.9%) | |
| No | 214(72.5%) | 81(27.5%) | 295(65.1%) | |
| **Perceived Stigma** | | | | <0.001* |
| Non- Stigmatized | 90 (83.3%) | 18(16.7%) | 108(23.8%) | |
| Stigmatized | 225(65.2%) | 120 (34.8%) | 345(76.2%) | |
| **Level of social support** | | | | 0.246 |
| Poor | 94 (64.4%) | 52(35.6%) | 146 (32.2%) | |
| Moderate | 79 (73.1%) | 29 (26.9%) | 108 (23.8%) | |
| Strong | 142(71.4%) | 57(28.6%) | 199 (43.9%) | |
| **Distance to the ART facility** | | | | 0.363 |
| ≤5 KM | 159 (72.6%) | 60 (27.4%) | 219 (48.3%) | |
| 5 to 10 KM | 88 (65.7%) | 46 (34.3%) | 134(29.6%) | |
| > 10 KM | 68 (68.0%) | 32 (32.0%) | 100(22.1%) | |
| Mean=7.23 KM (SD=3.677) | | | | |
| **Transportation to the ART facility** | | | | 0.174 |
| By foot | 147(65.9%) | 76(34.1%) | 223(49.2%) | |
| By bike | 49(69.0%) | 22(31.0%) | 71(15.7%) | |
| By car or taxi | 119(74.8%) | 40(25.2%) | 159(35.1%) | |
| **Use ART reminder** | | | | 0.007* |
| No | 130(63.1%) | 76(36.9%) | 206(45.5%) | |
| Yes | 185(74.9%) | 62(25.1%) | 247(54.5%) | |
| **Supportive family member** | | | | 0.004* |
| Have | 188(75.2%) | 62(24.8%) | 250(55.2%) | |
| Not have | 127(62.6%) | 76(37.4%) | 203(44.8%) | |
| **Smoking cigarettes** | | | | <0.001* |
| Yes | 41(49.4%) | 42(50.6%) | 83(18.3%) | |
| No | 274(74.1%) | 96(25.9%) | 370(81.7%) | |
| **IV drug abuse** | | | | 0.007* |
| Yes | 50(57.5%) | 37(42.5%) | 87(19.2%) | |
| No | 265(72.4%) | 101(27.6%) | 366(80.8%) | |
| **Khat chewing** | | | | 0.001* |
| Yes | 57(54.3%) | 48(45.7%) | 105(23.2%) | |
| No | 258(74.1%) | 90(25.9%) | 348(76.8%) | |
| **Multi sex partner** | | | | 0.847 |
| Yes | 15 (71.4%) | 6 (28.6%) | 21 (4.6%) | |
| No | 300 (69.4%) | 132 (30.6%) | 432 (95.4%) | |

*Significant level at a p-value <0.05

**Table 4. Clinical and laboratory characteristics between adherence and non-adherence.**

| Characteristics | ART adherence | | Total n (%) | p-value |
|---|---|---|---|---|
| | Adherence n (%) | Non-adherence n (%) | | |
| **CD4** (cell/mm³) | | | | 0.135 |
| <350 | 57 (62.0%) | 35(38.0%) | 92 (20.3%) | |
| 350-500 | 89 (74.8%) | 30 (25.2%) | 119 (26.3%) | |
| >500 | 169 (69.8%) | 73 (30.2%) | 242 (53.4%) | |
| **HB** (g/dl) | | | | 0.812 |
| ≤10 | 188(69.1%) | 84 (30.9%) | 272 (60.0%) | |
| >10 | 127(70.2%) | 54 (29.8%) | 181 (40.0%) | |
| **Regiment line** | | | | 0.352 |
| Second line | 76 (66.1%) | 39 (33.9%) | 115 (25.4%) | |
| First line | 239 (70.7%) | 99 (29.3%) | 338 (74.6%) | |
| **HIV Stage** | | | | 0.817 |
| Stage I | 185 (70.6%) | 77 (29.4%) | 262 (57.8%) | |
| Stage II | 80 (70.2%) | 34 (29.8%) | 114 (25.2%) | |
| Stage III | 26 (65.0%) | 14 (35.0%) | 40 (8.8%) | |
| Stage IV | 24 (64.9%) | 13 (35.1%) | 37 (8.2%) | |
| **HIV Viral load** (copies/mL) | | | | 0.024* |
| >1000 | 61 (60.4%) | 40 (39.6%) | 101 (22.3%) | |
| ≤1000 | 245 (72.2%) | 98 (27.8%) | 352(77.7%) | |
| **Years with HIV** | | | | 0.090 |
| < 2 | 77 (78.6%) | 21 (21.4%) | 98 (21.6%) | |
| 2–4 | 173 (67.1%) | 85 (32.9%) | 258 (57.0%) | |
| > 4 | 65 (67.0%) | 32 (33.0%) | 97 (21.4%) | |
| **Years with ART** | | | | 0.082 |
| < 2 | 64 (79.0%) | 17 (21.0%) | 81 (17.9%) | |
| 2–4 | 180 (66.2%) | 92 (33.8%) | 272 (60.0%) | |
| > 4 | 71 (71.0%) | 29 (29.0%) | 100 (22.1%) | |
| **ART side effect for the past 4 months** | | | | 0.028* |
| Yes | 52 (59.8%) | 35 (40.2%) | 87 (19.2%) | |
| No | 263 (71.9%) | 103 (28.1%) | 366 (80.8%) | |
| **Missed ART appointment for the past 4months** | | | | 0.058 |
| Yes | 40 (59.7%) | 27 (40.3%) | 67 (14.8%) | |
| No | 275(71.2%) | 111 (28.8%) | 386 (85.2%) | |
| **ART daily dosing** | | | | 0.001* |
| Twice daily | 53 (55.8%) | 42 (44.2%) | 95 (21.0%) | |
| One daily | 262 (73.2%) | 96 (26.8%) | 358 (79.0%) | |
| **Co-morbid disease** | | | | 0.003* |
| Yes | 169(64.0%) | 95(36.0%) | 264 (58.3%) | |
| No | 146 (77.2%) | 43 (22.8%) | 189 (41.7%) | |
| **History of OIS** | | | | <0.001* |
| Yes | 61 (54.5%) | 51 (45.5%) | 112 (24.7%) | |
| No | 254 (74.5%) | 87 (25.5%) | 341 (75.3%) | |

*Significant level at a p-value <0.05

Table 5. Factors associated with ART non-adherence in multivariable logistic regression analysis.

| Characteristics | AOR (95%CI) | p-value |
|---|---|---|
| **Cohabitation status** | | |
| Alone | 1.81(1.182-2.782) | 0.006* |
| With people | 1.00 | |
| **Perceived Stigma** | | |
| Non-stigma | 1.00 | |
| Stigma | 3.19(1.784-5.720) | < 0.001* |
| **Smoking cigarettes** | | |
| Yes | 2.96(1.759-5.003) | < 0.001* |
| No | 1.00 | |
| **Co-morbid disease** | | |
| Yes | 1.80 (1.155-2.813) | 0.010* |
| No | 1.00 | |

*Significant level at a p-value <0.05

studies' magnitudes of non-adherence to ART are lower than our observed result. The higher level of suboptimal non-adherences to ART among the Somali population living with HIV observed in this study might be connected to the nation's long-term conflict and natural disasters, which has led to significant displacement of civilians and increased economic vulnerability as well as disruption of preventive and curative health services.

Contrariwise, our study reveals a comparatively lower magnitude of non-adherence to ART when juxtaposed with findings from studies conducted in Cameroon [20], southern India [37], west Shewa, Ethiopia [38], and Rwanda [39]. These disparities may be attributed to socioeconomic variation, stigma and discrimination, different population profiles, geographical disparities, cultural or personal beliefs and misconceptions about HIV and its ART, healthcare system issues, ART accessibility, burden of mental health differences, and differences in sample sizes.

This study discovered that perceived stigma is significantly associated with ART non-adherence. Prior studies in Somalia have shown widespread HIV discrimination, stigma, depression, anxiety, and recurrent opportunistic infections, making it difficult for individuals living with HIV to stick to their ART [14–17]. A study conducted in the study area reported stigma, inadequate social support, sub-optimal ART adherence and substance abuse, as factors associated with depression among people living with HIV [17]. Effective HIV care goes beyond ART administration; it requires a holistic approach, considering various individual and social factors, particularly in this study area. This study underscores the importance of addressing not only the physical aspects of the disease but also providing comprehensive support, education, counselling, and material resources to navigate the broader context of HIV care, which ultimately promotes a sustainable HIV treatment to ensure optimal adherence and well-being for this vulnerable population.

This study found that individuals with co-morbid chronic diseases were 1.80 times more likely to be non-adherent to ART than those without such conditions. This can be attributed to the increased complexity of treatment, potential treatment fatigue, and drug interactions associated with managing multiple medications, appointments, and treatment plans, all of which can contribute to non-adherence to ART. Co-morbid cases hesitate to take medications with potential interactions since non-communicable disease has a high burden in the study area [40,41]. In addition, overlapping or exacerbating symptoms between co-morbid

conditions and HIV can cause patient confusion regarding the source of symptoms and reduce confidence in ART [42]. Furthermore, managing multiple health conditions can lead to psychological conditions like stress, anxiety, and depression, negatively impacting a person's motivation and ability to adhere to their treatment plans, including ART. Lastly, while ART medication is free of charge in the study area, treating co-morbid diseases can be costly, and individuals with co-morbidities may struggle to afford all the necessary medications and healthcare expenses, leading to missed doses of ART and subsequent non-adherence. Similar studies in Somalia and elsewhere supported [16,17,40–43].

The odds of not adhering to ART in this study were higher for individuals living alone than those living with others. This might be because cohabiting with a partner, family members, or roommates often has access to a built-in social support system. Cohabiting with others can provide emotional support, encouragement, and reminders to take ART regularly. Previous studies in the study area reported a high burden of poor social support with a widespread stigma, self-isolation, depression, and limited life opportunities. They report that the HIV response in Somalia is jeopardized by inadequate social support and discrimination [14–17]. Gokarn et al. [44] supported and reported that ART adherence was significantly associated with family or friends' reminders to take medications. On the other hand, Bomfim IG et al. [45] found that living alone and not being sexually active is associated with ART adherence. Besides, it is strongly advisable to enhance social support within this demoralized community, as doing so can promote treatment adherence by fostering motivation and responsibility among individuals for their prescribed treatment routines.

Finally, our study reveals that individuals who smoke cigarettes are more likely to be non-adherent to ART when compared to non-smokers. Psychosocially, smoking can lead to stress, resulting in suboptimal ART adherence, diminishing the quality of life by enhancing the risk of comorbid conditions, and can lead to depression [17,46–48]. In addition, it can exacerbate HIV-related health concerns, adversely interact with ART medications, and weaken the already compromised immune system, which can increase the risk of opportunistic infections. Similar studies have supported and reported that smoking is high among people living with HIV and is linked to suboptimal ART adherence, higher viral load, and missing doctor appointments [4–50]. Moreover, smoking worsens the severity and frequency of side effects and increases the risk of drug interactions, particularly in individuals with advanced HIV stages, ultimately leading to suboptimal ART adherence [51–54]. However, this study emphasizes addressing substance abuse issues to improve ART adherence and the overall health and well-being of people living with HIV in Somalia.

While this study boasts several commendable strengths, including being the first of its kind within the study area, it is vital to acknowledge its limitations. Being a single central study with a cross-sectional design, it cannot establish causal relationships between the examined factors and the observed outcomes.

## Conclusion

Individuals living with the human immunodeficiency virus (HIV) in Mogadishu, Somalia suffer sub-optimal adherence to antiretroviral therapy (ART). The primary reasons for ART non-adherence include forgetfulness, transportation difficulties, side effects, other commitments, HIV-related stigma, and distant ART facilities. The implementation efforts to reduce or eliminate non-adherence to ART must specifically address the needs of those living alone and individuals with co-morbid chronic conditions. In addition, reducing or eliminating perceived HIV stigma and discouraging substance abuse should be strategically prioritized to improve ART adherence, ultimately leading to an effective HIV response.

## Acknowledgment

The authors acknowledge the healthcare workers of the study hospital and respective participants for their cooperation and for providing necessary information.

## Author contributions

**Conceptualization:** Bashiru Garba, Pamornsri Inchon.

**Data curation:** Jamal Hassan Mohamoud.

**Formal analysis:** Abdirahman Khalif Mohamud.

**Investigation:** Abdirahman Khalif Mohamud, Najib Isse Dirie, Maryan Abdullahi Sh Nur.

**Methodology:** Abdirahman Khalif Mohamud.

**Project administration:** Mohamed Mustaf Ahmed.

**Resources:** Abdirahman Khalif Mohamud.

**Validation:** Abdirahman Khalif Mohamud.

**Visualization:** Abdirahman Khalif Mohamud, Maryan Abdullahi Sh Nur.

**Writing – original draft:** Bashiru Garba, Pamornsri Inchon.

**Writing – review & editing:** Bashiru Garba, Najib Isse Dirie, Pamornsri Inchon.

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
