## [Decision Letter · Decision Letter 0]

19 Dec 2023

PONE-D-23-36063Factors Associated with Non-Adherence to Antiretroviral Therapy (ART) in Adult Individuals Living with Human Immunodeficiency Virus (HIV) in Mogadishu, Somalia: A hospital-based Cross-Sectional StudyPLOS ONE

Dear Dr. Mohamud,

Thank you for submitting your manuscript to PLOS ONE. After careful consideration, we feel that it has merit but does not fully meet PLOS ONE’s publication criteria as it currently stands. Therefore, we invite you to submit a revised version of the manuscript that addresses the points raised during the review process.

**ACADEMIC EDITOR: **

We look forward to receiving your revised manuscript.

Kind regards,

Werku Etafa

Academic Editor

PLOS ONE

Reviewers' comments:

Reviewer's Responses to Questions

**Comments to the Author**

1. Is the manuscript technically sound, and do the data support the conclusions?

Reviewer #1: Yes

2. Has the statistical analysis been performed appropriately and rigorously? 

Reviewer #1: No

3. Have the authors made all data underlying the findings in their manuscript fully available?

Reviewer #1: Yes

4. Is the manuscript presented in an intelligible fashion and written in standard English?

Reviewer #1: No

5. Review Comments to the Author

Reviewer #1: Thank you for giving me the chance to review this paper. I found the manuscript dealt with one of the public health issues. This might provide health policymakers and healthcare providers with an important clue.

Some of the concerns are included below

1. General

Paper needs language and grammatical edition

The manuscript has no line number to mention the issue easily

2. Abstract

Why unnecessary abbreviations in the Abstract? Rewrite them in full statement since it is conclusion it reduces its attractiveness to reader, may be only AOR.

The keywords: your keywords were about nine, better if you minimize it to the standard and add full forms of shortened words or acronyms and abbreviations.

3. Methods

Better if you add the health profession profile data and level of ART clinic services from the study area in related to your study problem

4. Inclusion/exclusion criteria

The author excluded incomplete charts, what are these incomplete charts? Better if operationalized such words

5. Sample size determination

The author included 453 participants after calculating sample size which was 398. What is your evidence to use 453 as a sample size? If so what is the importance to calculate sample size by using the proportion of the problem? Let the author describe his reasons

6. Data collection procedure

The author used both primary and secondary data, how could you manage the recall bias since you asked the previous history of remembered to take pills? Or you should include in the limitation of the study

7. Data Analysis procedure

The author said the multivariate analysis was carried out in order to identify factors associated with ART non-adherence. Do you think there could be confounding factors? If so how did you manage them? Did you check the multicholinerity or VIF? I need clarification

8. Results

From the beginning what is the significance of this study? Let me hear your new finding

Look at your descriptive part it should be modified to as of more attractive for readers and better if you describe 2/3 of your variables included in the defined table.

9. Discussion

Your justification needs strong modifications

It was not clear your justification was for which part

Add the general finding of these references you tried to compare your result to be clear for the readers, now it is not clear whether it is really higher, lower or comparable with yours.

Discuss for comparable results, higher results and lower results in different and your justification should be also for each difference. Even I couldn’t find another studies justification or support for your some variables eg, perceived stigma. Rewrite them

Generally the discussion part needs strong justifications

10. Conclusion:

The author concluded his findings by high burden of non-adhering to ART but didn’t describe his baseline to say high, I think as the author report there was no study result in the country with the same problem. Please add your baseline why say high burden on non-adherence to ART

11. References

Some of your references are outdated (13, 24, 25, 26 ....) please revise them

As a major

The issue of your sample size

You focused only single institution

Didn’t measure the confounding factors

The way you write discussion

6. PLOS authors have the option to publish the peer review history of their article (what does this mean? ). If published, this will include your full peer review and any attached files.

**Do you want your identity to be public for this peer review?** For information about this choice, including consent withdrawal, please see our Privacy Policy .

Reviewer #1: No

---

## [Author Response · Author response to Decision Letter 1]

17 Jan 2024

For a detailed review, please refer to the attached point-by-point response document addressing all comments provided by the reviewers, as per your guidelines.

---

## [Decision Letter · Decision Letter 1]

13 May 2024

PONE-D-23-36063R1Factors Associated with Non-Adherence to Antiretroviral Therapy (ART) in Adult Individuals Living with Human Immunodeficiency Virus (HIV) in Mogadishu, Somalia: A hospital-based Cross-Sectional StudyPLOS ONE

Dear Dr. Mohamud,

Thank you for submitting your manuscript to PLOS ONE. After careful consideration, we feel that it has merit but does not fully meet PLOS ONE’s publication criteria as it currently stands. Therefore, we invite you to submit a revised version of the manuscript that addresses the points raised during the review process.

Please follow the journal guidelines regarding all matters relating to the manuscript.

Please submit your revised manuscript by Jun 27 2024 11:59PM. If you will need more time than this to complete your revisions, please reply to this message or contact the journal office at plosone@plos.org . Please include the following items when submitting your revised manuscript:

We look forward to receiving your revised manuscript.

Kind regards,

Werku Etafa

Academic Editor

PLOS ONE

Journal Requirements:

**Additional Editor Comments:**

Dear author (s),

Thank you for your effort to give feedback. Please, try to adhere to the journal guidelines for all the requirements to be included in the study. Please make the corrections in the manuscript, line by line and provide your feedback for this specific question one by one. 

Reviewers' comments:

Reviewer's Responses to Questions

**Comments to the Author**

1. If the authors have adequately addressed your comments raised in a previous round of review and you feel that this manuscript is now acceptable for publication, you may indicate that here to bypass the “Comments to the Author” section, enter your conflict of interest statement in the “Confidential to Editor” section, and submit your "Accept" recommendation.

Reviewer #1: All comments have been addressed

2. Is the manuscript technically sound, and do the data support the conclusions?

Reviewer #1: Yes

3. Has the statistical analysis been performed appropriately and rigorously? 

Reviewer #1: Yes

4. Have the authors made all data underlying the findings in their manuscript fully available?

Reviewer #1: Yes

5. Is the manuscript presented in an intelligible fashion and written in standard English?

Reviewer #1: No

6. Review Comments to the Author

Reviewer #1: Please revise the grammar more

Minimize the background from the abstract

Again, look at your discussion and conclusion

7. PLOS authors have the option to publish the peer review history of their article (what does this mean? ). If published, this will include your full peer review and any attached files.

**Do you want your identity to be public for this peer review?** For information about this choice, including consent withdrawal, please see our Privacy Policy .

Reviewer #1: No

---

## [Author Response · Author response to Decision Letter 2]

4 Nov 2024

A detailed, point-by-point response is separately attached.

---

## [Decision Letter · Decision Letter 2]

28 Feb 2025

Factors Associated with Non-Adherence to Antiretroviral Therapy (ART) in Adult Individuals Living with Human Immunodeficiency Virus (HIV) in Mogadishu, Somalia: A hospital-based Cross-Sectional Study

PONE-D-23-36063R2

Dear Dr. Abdirahman Khalif Mohamud,

We’re pleased to inform you that your manuscript has been judged scientifically suitable for publication and will be formally accepted for publication once it meets all outstanding technical requirements.

Kind regards,

Sana Eybpoosh

Academic Editor

PLOS ONE

Additional Editor Comments (optional):

Reviewers' comments:

Reviewer's Responses to Questions

**Comments to the Author**

1. If the authors have adequately addressed your comments raised in a previous round of review and you feel that this manuscript is now acceptable for publication, you may indicate that here to bypass the “Comments to the Author” section, enter your conflict of interest statement in the “Confidential to Editor” section, and submit your "Accept" recommendation.

Reviewer #1: All comments have been addressed

Reviewer #2: All comments have been addressed

2. Is the manuscript technically sound, and do the data support the conclusions?

Reviewer #1: Partly

Reviewer #2: Yes

3. Has the statistical analysis been performed appropriately and rigorously? 

Reviewer #1: Yes

Reviewer #2: Yes

4. Have the authors made all data underlying the findings in their manuscript fully available?

Reviewer #1: Yes

Reviewer #2: Yes

5. Is the manuscript presented in an intelligible fashion and written in standard English?

Reviewer #1: Yes

Reviewer #2: Yes

6. Review Comments to the Author

Reviewer #1: Didn't correct the abbreviation from the abstract (ART). What is ART? Write in full statement. I'm not satisfied with your sample size; what is your evidence to use this number?.

Reviewer #2: Dear authors,

Well done for revising this manuscript. It looks a lot better. I have no additional comments.

Thank you.

7. PLOS authors have the option to publish the peer review history of their article (what does this mean? ). If published, this will include your full peer review and any attached files.

**Do you want your identity to be public for this peer review?** For information about this choice, including consent withdrawal, please see our Privacy Policy .

Reviewer #1: No

Reviewer #2: No

---

## [Editor Report · Acceptance letter]

PONE-D-23-36063R2

PLOS ONE

Dear Dr. Mohamud,

I'm pleased to inform you that your manuscript has been deemed suitable for publication in PLOS ONE. Congratulations! Your manuscript is now being handed over to our production team.

Kind regards,

on behalf of

Dr. Sana Eybpoosh

Academic Editor

PLOS ONE